# Using a ResNet-18 Network to Detect Features of Alzheimer’s Disease on Functional Magnetic Resonance Imaging: A Failed Replication. Comment on Odusami et al. Analysis of Features of Alzheimer’s Disease: Detection of Early Stage from Functional Brain Changes in Magnetic Resonance Images Using a Finetuned ResNet18 Network. *Diagnostics* 2021, *11*, 1071

**DOI:** 10.3390/diagnostics12051094

**Published:** 2022-04-27

**Authors:** Peter J. Nicholas, Alex To, Onur Tanglay, Isabella M. Young, Michael E. Sughrue, Stéphane Doyen

**Affiliations:** Omniscient Neurotechnology, Sydney, NSW 2000, Australia; peter.nicholas@o8t.com (P.J.N.); alex.to@o8t.com (A.T.); onur.tanglay@o8t.com (O.T.); isabella.young@o8t.com (I.M.Y.); stephane.doyen@o8t.com (S.D.)

There is considerable interest in developing effective tools to detect Alzheimer’s Disease (AD) early in its course, prior to clinical progression. Among efforts is to predict the conversion of Mild Cognitive Impairment (MCI) to AD [1]. Although those with MCI are more likely to develop AD, many patients do not progress to dementia [2]. The unavailability of tools to screen and monitor patients makes it difficult to enroll suitable candidates for clinical trials, with subsequent failure of therapies which may require administration at an earlier stage of disease [3]. It is crucial to utilize data-driven methods relying on early structural and functional changes to improve clinical management of AD. We therefore followed with interest the evidence presented in *Diagnostics* by Odusami et al. [4] utilizing deep learning based-methods to predict MCI from AD and controls based on resting-state functional magnetic resonance imaging (rsfMRI) with near 100% accuracy, outperforming any previously reported prediction model. In this correspondence, we sought to provide an independent replication of this method, as establishing the reproducibility of these findings is imperative to facilitate their clinical translation.

We acquired the functional magnetic resonance imaging (fMRI) scans of 822 subjects from the Alzheimer’s Disease Neuroimaging Initiative 3 (ADNI-3) dataset (http://adni.loni.usc.edu/ (accessed on 1 October 2021)). For subjects who had multiple scan dates, the earliest scan was used to avoid duplicates. According to baseline diagnosis, 436 subjects were controls, 199 had Mild Cognitive Impairment (MCI), 88 had early MCI (EMCI), 37 had late MCI (LMCI), and 62 had Alzheimer’s Disease (AD). Considering the width, height, depth and time dimensions, the shape of each BOLD image was either (99, 117, 95, 197) or (99, 117, 95, 976), therefore generating either 18,715 or 92,720 slices. Twenty five subjects were randomly selected from each diagnostic group. Each rsfMRI image was sliced along the depth and time dimensions, and to only keep slices with prominent features, we calculated the mean BOLD signal of each slice and only kept slices with means in the 90th percentile for each subject. This filtering process manifested in around 1871 or 9270 slices per subject, and 700 slices were randomly selected from each subject for binary classification. The fMRI images were not further processed, and the images were enhanced in the way described in the original study to enable inputting of the dataset into the pretrained ResNet-18 model. 

We used the same neural net architecture described in the original study, ResNet-18, with a remodeling of the final dense layer to enable binary classification, and the addition of a non-linear ReLU activation function and a Dropout of 0.2. The layers were unfrozen to update the parameters of the pretrained model based on the new dataset. In order to allow for faster learning and utilize the large memory of our GPU, we also increased the batch size to 256 and learning rate to 1 × 10^−5^. For each binary classification, the data were split into training and test sets by subject: 17 subjects comprised the training set (11,900 slices per diagnosis) and 8 subjects the test set (5600 slices per diagnosis). The performance of each model was assessed using accuracy, specificity, and sensitivity, averaged over.

As seen in Table 1, we were unable to replicate the results reported in the previous study across any of the binary classifications. The models consistently performed worse than the replicated study. However, beyond this, performance was even worse than other attempts to screen for MCI or AD, using machine learning or neuropsychological testing, and therefore failing against any benchmark for clinical implementation. Whereas the original study reported an accuracy of 99.99%, 99.95%, and 99.95% when classifying EMCI vs. AD, LMCI vs. AD, and MCI vs. EMCI, respectively, our models achieved 45.24%, 44.12%, and 46.22% on the same. The highest accuracy was 56.97% when classifying CN vs. LMCI. In addition, both sensitivity and specificity were significantly lower than those reported in the original study, with the model attaining a less than 50% sensitivity and specificity on most classification problems. LMCI vs. AD had the lowest sensitivity at 2.62% and highest specificity at 52.56% Over all seven binary classification problems, the median accuracy, sensitivity, and specificity were 46.22%, 25.14%, and 33.00%, respectively (compared to 99.76%, 99.56%, and 100% in the original).

We subsequently experimented with an alternative holdout strategy. Instead of splitting the data into train and test sets by subject, we split them by fMRI slices along the axial axis, taking 70% of slices for training, and 30% of slices for testing. As a result, the train and test sets could have slices from the same subject. As seen in the performance metrics in Table 1 (Data Split by Slice), these classifications achieved significantly higher accuracy, specificity, and sensitivity than the first experiment, performing much closer to the metrics reported in the original study. These models still did not perform as well as those reported in the original study. The model with the highest accuracy was MCI vs. EMCI at 89.86%. This classification was also among those with highest accuracy in the original paper, but at a much higher 99.95%. This was only second to classifying EMCI vs. AD, with an accuracy of 99.99% in the original, whereas our model was only able to achieve 79.55% on the same. The original paper also reported that the proposed model achieved the highest accuracy, sensitivity, and specificity for CN vs. EMCI classification compared to other reported models. Upon replication, however, our model achieved an accuracy, sensitivity, and specificity of 86.90%, 85.75%, and 88.00%, respectively, when classifying CN vs. EMCI. Using this hold-out method attained a median accuracy, sensitivity, and specificity of 84.12%, 82.36%, and 88.47%, respectively.

It is therefore clear that splitting the data into train and test sets by fMRI slices resulted in significantly better performance. However, it is crucial to note that that this splitting method is invalid as it manifests in data leakage, whereby information from the test set leaks into the dataset used to train the model. The model is therefore likely recognizing the patients it has already encountered in the training set, rather than recognizing the features of MCI or AD. While these models may appear to perform successfully, if presented with data the model has never encountered, such as in our first replication experiment, performance would drop. In the real world however, a diagnostic problem would be similar to this first replication experiment where the model has not previously encountered the patient’s data and would need to be robust enough to classify the patient into a diagnostic category. Models with data leakage, therefore, provide a false sense of performance and are likely not robust enough for use in the clinical environment.

We herein demonstrate that a replication of the methods proposed by Odusami et al. fails to achieve the impressive performance reported by the authors. Although our results came close when using an invalid hold-out strategy, the original study reported that data were split by subjects rather than slices. We are therefore unable to identify the reasoning for the discrepancy between our results. While some level of discrepancy was expected given the known difficulty of replication with deep learning models, we are unable to explain the vastness of the difference reported. We were also unable to gain access to the original code used in the study, making it difficult to understand the reasoning behind the misclassification. The unavailability of the code also makes it difficult to be certain that our methods were an exact replica, but we aligned as closely as possible to the described methods in the paper. One difference between our methods and the original study was the utilization of the ADNI-3 dataset rather than ADNI-2. We used the ADNI-3 dataset as it had the 25 subjects per each diagnostic category without relying on repeat scans where subjects converted between categories longitudinally as in ADNI-2. We do not believe this would have impacted the validity of our replication, as the described method should be capable of replication in any similar dataset. The utility of a diagnostic tool should also not be dependent on the dataset used, as this speaks directly to the external validity of the tool in practice. Nonetheless, the stark contrast in results between the two hold-out strategies demonstrates the importance of using test sets which have never been encountered by the model and avoiding data leakage, which may ultimately compromise the applicability of the results.

The utilization of ResNet-18, including with the modifications proposed in the original paper, may also be too reductionistic for use in fMRI data, resulting in the misclassification results we have encountered. ResNet-18, or even other methods which have been applied to fMRI, ignores the time and depth dimensions of fMRI, treating the problem the same as any other image classification. This may however manifest in information loss, which may be crucial for detecting the features of pathology early in their course. Future studies should aim to utilize models incorporating the entire four dimensions of fMRI data in order to improve classification by considering the entirety of the information available.

Our results speak to the replication crisis in machine learning, which presents one of the foremost barriers to translating these methods to clinical practice. While models provide a fast and potentially accurate means of diagnosis, there are also difficulties with generalizing methods to new datasets, especially in cases where there is overfitting of the model to the training data, and proper precautions are not taken to prevent data leakage with consequent unreliable performance metrics. This remains one of the limitations of studies utilizing machine learning models, and a barrier slowing down clinical translation. Misclassification and reproducibility can, however, be improved in future studies if the field prioritizes translatability as much as, if not more than, novelty and outperformance. As echoed by Heil et al. [5], to prevent misclassification and improve reproducibility by other researchers, studies utilizing machine learning models in the life sciences must meet certain standards. Primarily, studies must accurately report the details of analysis including data-splitting procedures, but also make the data, models, and code publicly available to enable others to replicate the analysis and identify possible bias. Bringing the machine learning revolution to diagnostic medicine demands adherence to stringent data analysis and sharing practices and a prioritization of replication by independent groups on independent datasets.

## Figures and Tables

**Table 1 diagnostics-12-01094-t001:** Evaluation on test data utilizing two hold-out strategies when splitting the training and test sets.

	Data Split by Subject (Brain Hold-Out)			Data Split by Slice (Slice Hold-Out)		
Binary Classes	Accuracy (%)	Sensitivity (%)	Specificity (%)	Accuracy (%)	Sensitivity (%)	Specificity (%)
EMCI vs. LMCI	53.08	23.96	42.78	83.90	79.39	88.47
AD vs. CN	48.53	23.20	33.00	73.18	48.21	98.16
CN vs. EMCI	40.04	25.14	20.07	86.90	85.75	88.00
CN vs. LMCI	56.97	27.17	40.54	84.12	75.18	93.07
EMCI vs. AD	45.24	28.00	24.26	79.55	82.36	76.74
LMCI vs. AD	44.12	2.62	52.56	84.58	95.69	73.48
MCI vs. EMCI	46.22	27.51	23.76	89.86	86.43	93.33

## Data Availability

Data are available from the corresponding author upon reasonable request.

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
