# Peer review of "Using a ResNet-18 Network to Detect Features of Alzheimer’s Disease on Functional Magnetic Resonance Imaging: A Failed Replication. Comment on Odusami et al. Analysis of Features of Alzheimer’s Disease: Detection of Early Stage from Functional Brain Changes in Magnetic Resonance Images Using a Finetuned ResNet18 Network. Diagnostics 2021, 11, 1071"

_diagnostics, 2022, doi:10.3390/diagnostics12051094_

Round 1
Reviewer 1 Report
The reported issues about Alzheimer’s Disease are good, I have the following comments
1) Authors are suggested to follow standarad approach in paper writing for example Introduction followed by literature survey, proposed approach, description of dataset, Experiments, Results and Discussions.
2) Though the the proposed work is good, I would like to see the paper in the standarad format.
3) I suggest authors to include detailed investigation and analysis of experiments
4) Discuss the major advatages of the proposed method comapred to the existing methods and include the limiations and futre works of the proposed approach
5) All numbers in the table and plots should be discussed in detail.
6) Discuss misclassification results and show why misclassification is happening and how it can be prevented.
7) Provide more interpretation for features for example you can use t-SNE
Author Response
1) Authors are suggested to follow standard approach in paper writing for example Introduction followed by literature survey, proposed approach, description of dataset, Experiments, Results and Discussions.
We thank the reviewer for their suggestion. As this submission is however a reply to another manuscript which was published within this journal, and no original data is being presented, we would like to maintain its format as a letter. This is similar to the following submission which featured in this journal:
1. Nenna, R., et al., Comment on Jaworska, J. et al. Consensus on the Application of Lung Ultrasound in Pneumonia and Bronchiolitis in Children. Diagnostics 2020, 10, 935. Diagnostics, 2021. 11(1): p. 55.
2) Though the proposed work is good, I would like to see the paper in the standard format.
We thank the reviewer for their comment. As we have aforementioned in point 1, our paper did not include a new proposal or data. Since it was a replication and reply to another submission, we would prefer to maintain the paper in a Letter to the Editor format.
3) I suggest authors to include detailed investigation and analysis of experiments
We thank the reviewer for their suggestion. We have now included further detail on the discussion of the performance of our models in comparison to the original study we replicated, in addition to discussion on the results we obtained in the context of the generalizability of machine learning models. As our methodology aimed to replicate the original paper as closely as possible, we have attempted to align as closely to the original paper as much as possible.
4) Discuss the major advantages of the proposed method compared to the existing methods and include the limitations and future works of the proposed approach
We thank the reviewer for their comment. While the method we used in this study was aiming to replicate what was described by Odusami et al. in their original paper, and was not one we proposed for use in future studies. We have included some further discussion in regards to the advantages and disadvantages of machine learning approaches, their limitations and our opinion on future works in lines 134-150.
5) All numbers in the table and plots should be discussed in detail.
We thank the reviewer for their suggestion. We have included further detailed discussion on the performance of our replication models featured in Table 1 and compared these to the original study (lines 62-67, and lines 76-86). Our submission did not otherwise include any plots.
6) Discuss misclassification results and show why misclassification is happening and how it can be prevented.
We thank the reviewer for their comment. The latter part of our letter, starting from line 88 focuses on why we believe our models behaved differently to the original paper. We are however unable to provide further insight into this as the code used in the original study is not publicly available. We have also been denied access to the original code by the authors upon request. We can therefore only hypothesize into the behaviour of our study which attempted to replicate the described methods of the original paper. Nonetheless, we have made some modifications to lines 88-120 to make it more clear on why we believe misclassification is happening and how future models can prevent this.
7) Provide more interpretation for features for example you can use t-SNE
We thank the reviewer for their suggestion. Within this submission, we attempted to replicate the same methodology as the original paper as much as possible. We therefore did not include any additional analysis the original authors did not include in their work. Furthermore, as far as we are aware, in a deep learning paradigm such as this where fMRI images are the input, features are not manually derived and put into a machine learning algorithm. Instead, the multiple layers of the neural network extract progressively higher level features from the raw fMRI image. Consequently, we do not believe it possible to explore these with t-SNE. Regardless, our primary aim with our analysis was to be as close to the original paper as possible.
Round 2
Reviewer 1 Report
No further comments and I recommend Accept